# Development, implementation and validation of resource-stratified guidelines in low-income and middle-income countries: a scoping review protocol

Dylan Griswold [1,2,3] Sara Venturini,[1,3] Nancy Carney,[4] Andres M Rubiano,[1,5] Peter John Hutchinson [1,3] Angelos G Kolias [1,3]

For numbered affiliations see end of article.

**Correspondence to**
Dylan Griswold;
dylgris@gmail.com

## ABSTRACT

**Introduction** Nearly every field of medicine has some form of clinical practice guidelines. However, only within the past 5–10 years has the medical community acknowledged the need for well-developed guidelines tailored to the local healthcare needs and the resources available. In most low-income and middle-income countries (LMICs), healthcare workers depend on guidelines developed in high-income countries (HICs), yet many interventions validated in a HIC are ineffective when implemented in an LMIC. The variation in infrastructure, medical personnel, technology and environmental conditions exhibited in LMICs relative to HICs necessitates a careful appraisal of the evidence base used in clinical guideline recommendations. This review aims to map the use of resource-stratified guidelines across all fields of medicine. The review seeks to answer three questions for the identified guidelines: (1) what was the method of development, (2) have they been implemented and, if so, (3) have they been validated.

**Methods** The search strategy will aim to locate studies from inception to November 2021. An initial limited search of PubMed and Scopus was undertaken to identify articles on the topic. The text words contained in the titles and abstracts of relevant articles, and the index terms used to describe the articles were used to develop a full search strategy for PubMed and Scopus. This scoping review will be conducted in accordance with the Joanna Briggs Institute (JBI) methodology for scoping reviews. Data to be extracted from each study will include population characteristics of both developers and intended implementation population, medical specialty, validation status, method of guideline development, whether the study is consensus or evidence-based in addition to a summary of recommendations for practice.

**Ethics and dissemination** Ethical approval is not required for this review. The plan for dissemination is to publish review findings in a peer-reviewed journal.

## STRENGTHS AND LIMITATIONS OF THIS STUDY

⇒ To the best of our knowledge, this protocol provides a detailed description of the first scoping review on the development, implementation and validation of resource-stratified guidelines in low-income and middle-income countries.
⇒ The protocol adheres to the rigorous JBI methodology for scoping reviews.
⇒ The protocol is being conducted by a multidisciplinary team with extensive experience of conducting high-quality reviews.
⇒ Given the narrow specificity of the review, very few papers may be included in the final analysis.

## INTRODUCTION

Best-practice, evidence-based, clinical practice guidelines have become the gold standard for patient care. However, these guidelines are, by and large, developed by high-income countries (HICs). Thus, data extracted from clinical trials and adequate statistically controlled comparative studies used to develop the guidelines are based on the enrolment population. Of the 173 532 trials registered on the International Clinical Trials Registry Platform between 2005 and 2013, 85% were registered to recruit in HICs.[1] For the same period, 4.7% were registered to recruit in lower-middle-income countries (LMICs) and 0.8% in low-income countries. Thus, 85% of those studies registered for recruitment in HICs have a very different sample population relative to the same study carried out with a sample population enrolled from LMICs with social, cultural, genetic and infrastructure differences.[1] This difference is real and has implications for implementing guidelines developed for and by HICs in an LMIC. In most LMICs healthcare workers depend on guidelines developed in HICs, yet many interventions validated in a HIC are ineffective when implemented in an LMIC.[2 3] The variation in infrastructure, medical personnel, technology and

environmental conditions exhibited in LMICs relative to HICs necessitates a careful appraisal of the evidence base used in clinical guideline recommendations.

Nearly every field of medicine has some form of clinical practice guidelines.[4] Nevertheless, only within the last 5–10 years has the literature acknowledged the need for well-developed guidelines tailored to the local healthcare needs and the resources available. The WHO has developed guidelines for different topics ranging from antimicrobial resistance and neglected tropical disease to environmental health and health systems.[5] Mock *et al* were renegades in this regard, publishing Trauma Mortality Patterns in Three Nations at Different Economic Levels: Implications for Global Trauma System Development—in 1998.[6] The outcome of the study showed that in all three cities, the majority of deaths occurred in the prehospital setting and that improving emergency care is especially vital in middle-income nations that have already established essential emergency medical services. Mock then led an initiative between the International Association for the Surgery of Trauma and Surgical Intensive Care and the WHO to establish the Working Group for Essential Trauma Care. The group sought to formulate a plan to address the difficulties that confront trauma care in LMICs. In 2004, the group published the document, Guidelines for essential trauma care, which set forth a list of essential trauma services in 14 categories of trauma care that the group felt was achievable in nearly every worldwide setting and included the human and physical resources that would ensure their provision.[7]

Though guidelines like those developed by Mock's team are available, they are situational, for which there are no comprehensive guidelines for each of the 14 categories of trauma care.

WHO's current trauma guidelines recognise that some protocols developed by HICs require the most costly resources in the 'modern therapeutic armamentarium', and that, 'it is unlikely that low-income or even middle-income countries will be able to meet these guidelines fully'.[7] Thus, the WHO's guidelines list the most essential diagnostic and therapeutic capabilities from HIC trauma guidelines. These recommendations outline essential procedures that the vast majority of LMIC facilities can perform. However, LMICs cannot implement these as official trauma guidelines as they fail to provide recommendations stratified by the resources available across multiple domains and levels of care for severe trauma.

One model that has demonstrated success is the resource-stratified guidelines developed for oncology care in areas with varying levels of resource availability. The Breast Health Global Initiative (BHGI) was organised in 2002 to improve the care of women with breast cancer in low-resource settings.[8] Their initiative was successful in developing clinical practice guidelines that account for varying levels of resources available in differing regions. Their methodology provides a framework for diagnosis and treatment recommendations across four resource levels: basic, limited, enhanced and maximal.[9–13] In recent years, the National Comprehensive Cancer Network (NCCN) have taken their evidence-based, consensus-driven clinical practice guidelines (NCCN Guidelines) developed for high-resource settings and have developed a framework for resource-stratifying the guidelines for implementation in resource-limited settings. They have formally designated this framework as the NCCN Framework for Resource Stratification of the NCCN Guidelines (NCCN Framework).[14] Like the BHGI guidelines, the NCCN Framework also identifies four resource environments: basic resources, core resources, enhanced resources and NCCN Guideline assumed resources. The current library of guidelines includes 16 different types of cancers in addition to breast cancer screening, adult cancer pain and palliative care.[15] According to 2016 data,[14] 47% of the 754 000 verified users of the internet-based guidelines were from 198 countries outside the USA.[14]

While the NCCN Framework has led to the development of a variety of cancer-specific resource-stratified guidelines, they have yet to be translated across other fields of medicine. The American Society of Clinical Oncology and the Asian Oncology Society have also developed their own set of cancer-specific resource-stratified guidelines.[16 17] It is largely unknown if and to what degree other fields of medicine have similar frameworks. Understanding how other fields of medicine have developed and implemented resource-stratified guidelines can inform development of new resource-stratified guidelines for specialities without them. Therefore, this review aims to map the use of resource-stratified guidelines across all fields of medicine. A preliminary search of PROSPERO, MEDLINE, the Cochrane Database of Systematic Reviews and the JBI Database of Systematic Reviews and Implementation Reports was conducted, and no current or underway systematic reviews on the topic were identified.

The findings of this review will inform stakeholders about relevant models to develop, implement and evaluate resource-stratified-guidelines in low-resource settings.

## METHODS

The proposed scoping review will be conducted in accordance with the JBI methodology for scoping reviews.[18]

### Review question

Question: Of the resource-stratified guidelines being used in medicine, what are the methods of development, have they been successfully implemented and, if so, have they been validated? Validation refers to whether the performance of the guidelines was assessed and, if so, the results from the assessment.

Objective: To identify resource-stratified guidelines being used in medicine and assess their methods of development, implementation and validation.

### Patient and public involvement

Patients and the public were not involved in the design of this scoping review protocol.

## Data source and search strategy

The search strategy will aim to locate both published and unpublished studies from inception to November 2021. An initial limited search of PubMed and Scopus was undertaken to identify articles on the topic. The text words contained in the titles and abstracts of relevant articles, and the index terms used to describe the articles were used to develop a full search strategy for PubMed and Scopus. A full search strategy for PubMed and Scopus is detailed in online supplemental appendix I. In the second phase of the search, a final search strategy will be adopted for each information source. The reference lists of all selected studies will be screened for additional studies during the third phase of the search. Unpublished studies will be captured through scanning societal meetings for abstracts and consensus-based guideline development conferences.

## Selection of studies

Following the search, all identified citations will be collated and uploaded into EndNoteX9 (Clarivate Analytics, Pennsylvania, USA). The citations will then be imported into Covidence online software (Veritas Health Innovation, Melbourne, Australia) for screening. Two independent researchers will examine titles and abstracts for inclusion. The full text of selected studies will be retrieved and assessed. Full-text studies that do not meet the inclusion criteria will be excluded, and the reasons for exclusion will be provided in an online supplemental appendix in the final scoping review. Any disagreements that arise between the researchers during either title and abstract screening or full-text screening will be resolved through discussion, or with a third reviewer. The results of the search will be reported in full in the final article and presented in a Preferred Reporting Items for Systematic Reviews and Meta-Analyses extension for Scoping Reviews (PRISMA-ScR) checklist.

## Eligibility criteria

Inclusion criteria.

## Participants

This scoping review will consider guidelines that were developed for low-income and middle-income countries, as defined by world bank data for medical specialties recognised by the European Union and European Economic Area.

## Concept

The concept of interest for the proposed scoping review is resource-stratified guidelines developed for low-income and middle-income countries across all fields of medicine. This will include, but not be limited to, population, medical specialty, intervention, validation status, method of guideline development, whether the study is consensus or evidence-based in addition to the comparator and outcome measures.

**Table 1** PCC inclusion criteria

| Participants | Guidelines developed for low-income and middle-income countries across all fields of medicine |
|---|---|
| Concept | Development, implementation and validation of resource-stratified guidelines |
| Context | Limited to guidelines developed for low-income and middle-income countries, as defined by world bank data for medical specialties recognised by the European Union and European Economic Area |

## Context

The review will be limited to guidelines developed for low-income and middle-income countries, as defined by world bank data for medical specialties recognised by the European Union and European Economic Area.

## Types of studies

This scoping review will consider both experimental and quasiexperimental study designs including randomised controlled trials, non-randomised controlled trials, before and after studies and interrupted time-series studies. In addition, analytical observational studies, including prospective and retrospective cohort studies, case-control studies and analytical cross-sectional studies, will be considered for inclusion.

Studies published in English and English abstracts of foreign language studies will be included. All studies will be considered regardless of publication date.

The Participants, Concept, Context (PCC) inclusion criteria is summarised in table 1.

## Data extraction

Data will be extracted from the papers included in the review by two independent researchers using the data extraction instrument (online supplemental appendix II). The following information will be extracted from the articles: (1) study title; (2) author; (3) population characteristics of guideline developers; (4) intended population for implementation; (5) rationale for development; (6) methodology; (7) development; (8) implementation; (9) validation; (10) summary of findings; (11) medical specialty; (12) practice implications; (13) recommendations for further development.

The draft data extraction tool will be modified and revised as necessary during the process of extracting data from each included study. Modifications will be detailed in the full scoping review report. Any disagreements that arise between the reviewers will be resolved through discussion, or with a third reviewer. Authors of papers will be contacted to request missing or additional data, where required.

## Data presentation

The extracted data will be presented in tabular form and as a narrative summary that aligns with the aim of this scoping review. The table will report: (i) distribution of studies by countries of guideline origin and intended population; (ii) methodology of development; (iii)

implementation and validation and (iv) summary of recommendations. Graphical representations may be used, including bar charts, line charts, pie charts and diagrams.

## Ethics and dissemination

No ethical approval will be required, as this review is based on already published data and does not involve interaction with human subjects. The plan for dissemination, however, is to publish the findings of the review in a peer-reviewed journal and present findings at high-level international conferences that engage the most pertinent stakeholders.

## DISCUSSION

This protocol has been rigorously developed and designed specifically to determine the methods of development, implementation and validation of resource-stratified guidelines being used across all fields of medicine. The findings of this review will inform stakeholders about relevant models to develop, implement and evaluate resource-stratified guidelines in low-resource settings.

## Limitations

This study will be limited by the relative novelty of guideline development for LMICs. Based on our initial search, there are very few examples of resource-stratified guidelines. However, the purpose of the review is to identify those that do exist and to extract the elements that will aid in future development of resource-stratified guidelines in LMICs.

**Author affiliations**
[1]NIHR Global Health Research Group on Neurotrauma, University of Cambridge, Cambridge, UK
[2]School of Medicine, Stanford Medical School, Stanford, CA, USA
[3]Department of Clinical Neurosciences, University of Cambridge, Cambridge, UK
[4]Department of Medical Informatics and Clinical Epidemiology, Oregon Health & Science University, Portland, Oregon, USA
[5]Neurosciences Institute, Universidad El Bosque, Bogota, Colombia

**Acknowledgements** The authors would like to thank Isla Kuhn, Head of Medical Library Services, University of Cambridge Medical Library for her assistance in developing the search strategy.

**Contributors** DG, AGK, PJH and AMR conceived the review. NC and SV helped with the development of the methods. DG drafted the manuscript and all authors contributed to its revision.

**Funding** This work is supported by the NIHR Global Health Research Group on Neurotrauma, which was commissioned by the National Institute for Health Research (NIHR) using UK aid from the UK Government (project 16/137/105). The views expressed in this publication are those of the author(s) and not necessarily those of the NIHR or the Department of Health and Social Care. DG is supported by the Gates Cambridge Trust (OPP1144). PJH is supported by a Research Professorship from the NIHR, the NIHR Cambridge Biomedical Research Centre, a European Union Seventh Framework Program grant (CENTER-TBI; grant no. 602,150) and the Royal College of Surgeons of England. AGK is supported by a Clinical Lectureship, School of Clinical Medicine, University of Cambridge and the Royal College of Surgeons of England.

**Competing interests** None declared.

**Patient and public involvement** Patients and/or the public were not involved in the design, or conduct, or reporting, or dissemination plans of this research.

**Patient consent for publication** Not applicable.

**Ethics approval** Not applicable.

**Provenance and peer review** Not commissioned; externally peer reviewed.

**Data availability statement** Data sharing not applicable as no datasets generated and/or analysed for this study.

**ORCID iDs**
Dylan Griswold http://orcid.org/0000-0003-0291-8360
Peter John Hutchinson http://orcid.org/0000-0002-2796-1835
Angelos G Kolias http://orcid.org/0000-0003-3992-0587

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
