## [Reviewer comments · BMJ Open]

ARTICLE DETAILS

TITLE (PROVISIONAL)	Development, Implementation, and Validation of Resource Stratified Guidelines in Low-and Middle-Income Countries: A Scoping Review Protocol
AUTHORS	Griswold, Dylan; Venturini, Sara; Carney, Nancy; Rubiano, Andres M; Hutchinson, Peter; Kolas, Angelos

VERSION 1 – REVIEW

REVIEWER	Wilson, Brooke Princess Margaret Hospital, Medical oncology
REVIEW RETURNED	01-Feb-2022

GENERAL COMMENTS	This is a clear and well written protocol on a topic of growing importance in medicine. The following minor suggestions may help to further strengthen the protocol. 1. In the introduction, the authors state that studies registered in high income countries have sample populations with biological homogeneity (line 12). I'm not convinced that simply being conducted in a high-income county leads to biological homogeneity. Even in high income countries, there can be significant racial and cultural heterogeneity which could contribute to biological heterogeneity and there can also be significant economic heterogeneity. Consider rephrasing. I think the real point is one of resource homogeneity, although even this may vary within countries.2. Within the introduction, it may also be useful to make reference to the following:a. Referencing the WHO list of essential medications as another example of priority setting and resource stratification for medicineb. The authors reference the BHGI and the NCCN as examples of RSG in oncology. It may be worth highlighting that there are an additional 2 cancer organisations producing RSG (ASCO and Asian Oncology Society). Moreover, are the authors aware of any organisations outside of oncology who have developed RSG?3. Based on my experience, I think the authors will find that there is very little information on guideline implementation and validation in LMICs. Can the authors be more specific in explaining how they will evaluate the implementation and validation of any identified guidelines? Much of this is often published in grey literature and is not easy to identify through systematic searches.4. In the methods, can the authors specify the date ranges of the searches? Looking at the appendix I believe this is from inception to November 2021?5. Do the authors plan to evaluate the "quality" of any identified guidelines? Arguably, this will be easier to assess than
--

	implementation and validation. This could be done using validated tools for guideline analysis such as the AGREE II questionnaire. 6. The following additional points for data extraction could be important to incorporate: a. Is the guideline developed for use in an individual country? Or as a reference for all countries of a given resource level (eg. All LIC or all LMIC). b. What is the income group of those developing the guidelines? (for example the NCCN RSG for cancer care have limited representation from clinicians in LMICs on the panel of members making the recommendations). I think this is very important to collect. 7. The authors state in the methods that they will “aim to locate both published and unpublished studies” but do not provide any details on how unpublished studies will be identified. Do they mean only abstracts? Or research published through other means (such as government policy documents or societal guidelines and position pieces). Do the authors plan to search the websites and policy documents of any major medical societies? Or specific countries? Please consider how best to capture these potential sources. If they will not be captured, this is an important limitation that should be highlighted.
--	--

REVIEWER	Owolabi, M. O. University of Ibadan College of Medicine, Medicine
REVIEW RETURNED	05-Feb-2022

GENERAL COMMENTS	It is a great idea to examine the concept of stratified guidelines , to check if it is appropriate and if it works. However because all end users deserve the best and not second or third rated interventions , a better idea could be to examine whether guidelines devote sufficient content to issues of implementation, contextualization, communication to all stakeholders and toolbox for amplification of facilitators/enablers and navigation of barriers in various settings. Moreover, there are low income settings in high income countries and high income settings in low income countries. The publications to be reviewed are guidelines. It is not clear why the following types of studies are listed "This scoping review will consider both experimental and quasi-experimental study designs including randomized controlled trials, non-randomized controlled trials, before and after studies, and interrupted time-series studies. In addition, analytical observational studies, including prospective and retrospective cohort studies, case-control studies, and analytical cross-sectional studies, will be considered for inclusion."
---

VERSION 1 – AUTHOR RESPONSE

Reviewer: 1
Dr. Brooke Wilson, Princess Margaret Hospital

Comments to the Author:

This is a clear and well written protocol on a topic of growing importance in medicine. The following minor suggestions may help to further strengthen the protocol.

1. In the introduction, the authors state that studies registered in high income countries have sample populations with biological homogeneity (line 12). I'm not convinced that simply being conducted in a high-income country leads to biological homogeneity. Even in high income countries, there can be significant racial and cultural heterogeneity which could contribute to biological heterogeneity and there can also be significant economic heterogeneity. Consider rephrasing. I think the real point is one of resource homogeneity, although even this may vary within countries.

We thank you for pointing this out. We have removed that phrasing.

2. Within the introduction, it may also be useful to make reference to the following:

a. Referencing the WHO list of essential medications as another example of priority setting and resource stratification for medicine

Thank you for this suggestion. We chose not to reference the list of essential medications as it is a broad list that is not specific to a particular field or specialty, which is our focal point.

b. The authors reference the BHGI and the NCCN as examples of RSG in oncology. It may be worth highlighting that there are an additional 2 cancer organisations producing RSG (ASCO and Asian Oncology Society). Moreover, are the authors aware of any organisations outside of oncology who have developed RSG?

The authors thank you for this suggestion. We have highlighted the RSGs of the other two cancer organizations. We are not aware of any other organizations outside of oncology who have developed RSGs. If there are, we hope to discover them in this scoping review.

3. Based on my experience, I think the authors will find that there is very little information on guideline implementation and validation in LMICs. Can the authors be more specific in explaining how they will evaluate the implementation and validation of any identified guidelines? Much of this is often published in grey literature and is not easy to identify through systematic searches.

The authors thank you for this suggestion. This is a component that is subject to be specific to the guidelines identified, and we are documenting the process of implementation and validation if mentioned in the identified study.

4. In the methods, can the authors specify the date ranges of the searches? Looking at the appendix I believe this is from inception to November 2021?

The authors thank you for pointing this out. We have included this detail in the methods section.

5. Do the authors plan to evaluate the "quality" of any identified guidelines? Arguably, this will be easier to assess than implementation and validation. This could be done using validated tools for guideline analysis such as the AGREE II questionnaire.

The authors thank you for this inquiry. We do not intend to evaluate the quality identified guidelines. This may be the topic of a future study.

6. The following additional points for data extraction could be important to incorporate:

a. Is the guideline developed for use in an individual country? Or as a reference for all countries of a given resource level (eg. All LIC or all LMIC).

b. What is the income group of those developing the guidelines? (for example the NCCN RSG for cancer care have limited representation from clinicians in LMICs on the panel of members making the recommendations). I think this is very important to collect.

The authors thank you for raising this point. This data will be included in the population characteristics of guideline developers and the intended population for implementation.

7. The authors state in the methods that they will “aim to locate both published and unpublished studies” but do not provide any details on how unpublished studies will be identified. Do they mean only abstracts? Or research published through other means (such as government policy documents or societal guidelines and position pieces). Do the authors plan to search the websites and policy documents of any major medical societies? Or specific countries? Please consider how best to capture these potential sources. If they will not be captured, this is an important limitation that should be highlighted.

Thank you for pointing this out. We have added, “Unpublished studies will be captured through scanning societal meetings for abstracts and consensus-based guideline development conferences.”

Reviewer: 2

Dr. M. O. Owolabi, University of Ibadan College of Medicine

Comments to the Author:

It is a great idea to examine the concept of stratified guidelines, to check if it is appropriate and if it works. However, because all end users deserve the best and not second or third rated interventions, a better idea could be to examine whether guidelines devote sufficient content to issues of implementation, contextualization, communication to all stakeholders and toolbox for amplification of facilitators/enablers and navigation of barriers in various settings.

Moreover, there are low income settings in high income countries and high income settings in low income countries.

The publications to be reviewed are guidelines. It is not clear why the following types of studies are listed "This scoping review will consider both experimental and quasi-experimental study designs including randomized controlled trials, non-randomized controlled trials, before and after studies, and interrupted time-series studies. In addition, analytical observational studies, including prospective and retrospective cohort studies, case-control studies, and analytical cross-sectional studies, will be considered for inclusion. "

The authors thank Dr. Owolabi for his comments. We agree that it is important to examine whether guidelines devote sufficient content to issues of implementation, contextualization, and communication to all stakeholders, and this may very well be the topic of a future paper. We first need to identify what resource stratified guidelines, if any, exist, before a paper of that significance can be written.

The types of studies to be included are listed so as to not miss any papers that reference and/or discuss resource-stratified guidelines.

VERSION 2 – REVIEW

REVIEWER	Wilson, Brooke Princess Margaret Hospital, Medical oncology
REVIEW RETURNED	20-May-2022
GENERAL COMMENTS	The authors have adequately addressed all of my concerns, and is suitable for publication. Thank you for the opportunity to review this work.

VERSION 2 – AUTHOR RESPONSE

Reviewer: 1

Dr. Brooke Wilson, Princess Margaret Hospital

Comments to the Author:

The authors have adequately addressed all of my concerns, and is suitable for publication. Thank you for the opportunity to review this work.

The authors would like to thank Dr. Brooke Wilson for taking the time to review our manuscript